# The Role of Extracellular Vesicles in Diseases of the Ear, Nose, and Throat

**DOI:** 10.3390/medsci11010006

**Published:** 2022-12-28

**Authors:** Jonathan M Carnino, Steven Miyawaki, Sanjeev Rampam

**Affiliations:** 1Department of Otolaryngology, Boston University School of Medicine, Boston, MA 02118, USA; 2Boston University School of Medicine, Boston, MA 02118, USA; 3Department of Orthopedic Surgery, Boston University School of Medicine, Boston, MA 02118, USA

**Keywords:** extracellular vesicle (EV), ear, nose, throat (ENT), otolaryngology

## Abstract

Extracellular vesicles (EVs) are membranous nanoparticles produced by most cell types into the extracellular space and play an important role in cell-to-cell communication. Historically, EVs were categorized based on their methods of biogenesis and size into three groups: exosomes, microvesicles, and apoptotic bodies. Most recently, EV nomenclature has evolved to categorize these nanoparticles based on their size, surface markers, and/or the cell type which secreted them. Many techniques have been adopted in recent years which leverage these characteristics to isolate them from cell culture media and biological fluids. EVs carry various “cargo”, including DNA, RNA, proteins, and small signaling molecules. After isolation, EVs can be characterized by various methods to analyze their unique cargo profiles which define their role in cell-to-cell communication, normal physiology, and disease progression. The study of EV cargo has become more common recently as we continue to delineate their role in various human diseases. Further understanding these mechanisms may allow for the future use of EVs as novel biomarkers and therapeutic targets in diseases. Furthermore, their unique cargo delivery mechanisms may one day be exploited to selectively deliver therapeutic agents and drugs. Despite the growing research interest in EVs, limited studies have focused on the role of EVs in the diseases of the ear, nose, and throat. In this review, we will introduce EVs and their cargo, discuss methods of isolation and characterization, and summarize the most up-to-date literature thus far into the role of EVs in diseases of the ear, nose, and throat.

## 1. Introduction

In the last decade, the role of EVs in various disease etiologies has been further studied and understood, most notably in the areas of cancer, infectious diseases, pulmonary diseases, cardiovascular diseases, and neurodegenerative diseases. Despite their rapidly emerging significance and interest, limited reports have investigated the role of EVs in diseases of the ear, nose, and throat (ENT). EVs have the potential to serve as novel diagnostic biomarkers which can be collected non-invasively and as a possible method of delivering therapeutics to target cells and organs. Furthermore, the ease of collecting biological fluids (saliva, sputum, ear discharge, etc.) from patients with ENT diseases makes the isolation of EVs relatively simple. In addition, using EVs therapeutically for ENT diseases would be simple given the ability to deliver them non-invasively to these patients. These reasons make EVs highly valuable to the field of otolaryngology. In this review, we will cover the biology, isolation methods, and characterization techniques of extracellular vesicles, while also summarizing the current literature which details the role and significance uncovered thus far to diseases of the ear, nose, and throat. 

## 2. Biology of Extracellular Vesicles

### 2.1. EV Biogenesis

Extracellular vesicles (EVs) are membranous nanoparticles produced by most cell types into the extracellular space and play an important role in cell-to-cell communication [1,2]. EVs contain “cargo” which is selectively packaged with the unique profiles of nucleic acids, proteins, and signaling molecules based on the status of the cell by which they are generated [2,3]. Prior to the 2018 guidelines set by the International Society of Extracellular Vesicles (ISEV), EVs were categorized into three different subtypes, and this older system of classification is often still used today. Given the lack of adoption of the newer nomenclature and the historical use of the three-category nomenclature used in older studies described below, we must introduce both systems of classification for clarity. Historically, EVs were categorized based on their size, biogenesis, and surface markers as exosomes (Exos), microvesicles (MVs), and apoptotic bodies (ABs) [2,4]. Exosomes are formed by the endosomal pathway, specifically the fusion of multivesicular bodies (MVBs) with the plasma membrane, and they usually range from 30–100 nm in diameter [5,6]. Microvesicles are formed by the outward budding and pinching off from the plasma membrane and typically measure from 100–1000 nm in diameter [7]. Finally, apoptotic bodies are produced by cells undergoing apoptosis, and they produce vesicles which are typically 1000–5000 nm in diameter [8]. The lack of agreement on surface markers for EVs described by this three-category system, in addition to the partial overlap on their size, has led to the development of new nomenclature for EV classification [9,10]. Updated guidelines established by the ISEV suggests categorizing EVs based on: (1) physical characteristics such as size (small, medium, or large EVs; with defined ranges) or density (low, middle, or high density; with defined ranges), (2) surface markers or biochemical compositions (Annexin A5-stained EVs, CD63+/CD81+ EVs, HSP70+ EVs, etc.), or (3) cell condition or origin cell type (hypoxic EVs, OSC19 EVs, apoptotic bodies, etc.) [9,10]. (Figure 1).

### 2.2. EV Cargo

EVs carry various types of cargo, such as proteins, nucleic acids, and lipids, and consequently act as important regulators in intercellular communication. Depending on the type, the bioactive cargo is sorted for EV transport through various mechanisms. Proteins are marked for EV transport through post-translational modifications (PTMs) such as ubiquitination, SUMOylation, or Neddylation. These ubiquitin-like PTMs allow for interaction of the protein with the endosomal sorting complex required for the transport (ESCRT) [11]. The ESCRT pathway is activated when the ubiquitinylated protein appears in early endosomes, after which the ESCRT complexes work cooperatively to sort ubiquitinylated protein into exosomes [12]. Proteins may also be sorted into EVs via lipid rafts, which are cholesterol and sphingolipid-rich structures that exist on the plasma membrane. The transfer of proteins to the raft allows for the formation of early exosomes [13]. In addition, tetraspanins play a role in protein sorting. CD63, for instance, is enriched on the intraluminal vesicles in late endosomes. It plays a critical role in the transporting of latent membrane protein 1 (LMP1) into EVs of cells infected with the Epstein–Barr Virus (EBV) [14]. 

EVs carry nucleic acids, such as mRNAs, DNAs, and miRNAs. RNA-binding proteins (RBPs) regulate the sorting of miRNAs into exosomes. For instance, miRNA with specific motifs bind to hnRNPA2B1, which then binds to the ceramide-rich membrane region of the MVB membrane to package miRNAs to the exosome [15]. The mRNA is identified for EV transport through sequence recognition of 3′UTR in the mRNA transcript by the ribonuclear protein complex. The presence of a CTGCC core domain in the 3′UTR promotes the loading of mRNA into EVs [16]. Other RBPs, such as SYNCRIP, YBX-1, MVP, FMR1, and AGO2, have also been found to play roles in RNA sorting into EVs [12]. EVs also carry DNA, and the presence of genomic DNA (gDNA) in EVs is a marker of genomic instability. In times of genomic instability, micronuclei form and often break down, after which CD63 surrounds their nuclear contents and loads them into exosomes [17].

Lastly, EVs often carry lipid cargo. Current research on lipid sorting mechanisms is limited, although the existing literature suggests that lipid sorting into EVs is related to the size and quantity of the exosomes [18].

### 2.3. EV Isolation

Ultracentrifugation is the most popular primary isolation method for EVs [19]. The conventional method of EV isolation is differential centrifugation. This technique separates and removes contaminants from EVs via several centrifugation steps. The major advantages of this technique include low processing cost and high processing quantity [20]. Density gradient centrifugation is another type of ultracentrifugation technique useful for the isolation of EVs. This method separates EVs into distinct layers based on their buoyant density. Lack of contamination with viral particles and pure preparation are major advantages of density gradient centrifugation [21]. However, there are still drawbacks to ultracentrifugation methods. The time-consuming process, expensive equipment, and dependence on rotary-tube type are the main challenges in using these techniques [21].

Other size-based methods for EV isolation include size-exclusion chromatography and ultrafiltration. In size-exclusion chromatography, a solution is passed through a column of porous beads smaller than the EVs to be isolated which separate biomolecules based on their hydrodynamic radii [2]. The main advantages of this technique include deterring EV aggregation and minimal damage to the EV [22]. A challenge with this technique is the possibility of molecules similar in size to the EV of interest contaminating the eluted product [2]. Porous membranes are used in ultrafiltration to capture molecules of a specific size and elute smaller molecules. For the isolation of EVs, ultrafiltration is typically performed in successive steps [20]. Notable advantages for this technique include the ability for the simultaneous processing of many samples and the absence of constraints on sample volume [21]. While faster and less laborious than other methods, sample contamination and loss of sample due to filter plugging are major obstacles in using this technique [20].

Affinity-based techniques target EV surface protein markers including CD63, CD8, and CD9 [23]. Specific antibodies with affinity to these surface markers are attached to magnetic beads. The sample solution is then typically run through these magnetic beads to capture EVs which are later washed out [24]. Centrifugation techniques are most often paired with this method to further isolate EVs after centrifugation. The high purity of isolated EVs and high selectivity of the antibodies are major advantages for affinity-based techniques. Drawbacks to these techniques include difficulty with characterization of non-intact vesicles, difficulty in washing out EVs after capture, and high cost.

Precipitation-based techniques modify the solubility of EVs to isolate them into a pellet. Precipitation of EVs can be performed using organic solvents, sodium acetate, protamine, or polyethylene glycol (PEG) [25]. Organic solvents allow for an ion-pairing effect that can provide high efficiency for the precipitation of EVs from these solutions [26]. PEGs are often used in commercial kits for EV isolation and result in the precipitation of EVs out of the solution into a pellet. This pellet can be analyzed further, but it often contains other contaminants with similar solubility. Sodium acetate’s utility as a precipitation solvent derives from its ability to aggregate EVs by the hydrophobic effect to precipitate a pellet out of the solution. The positively charged protamine solution can interact with and aggregate the negatively charged EVs [27]. Procedural simplicity, low cost, and high sample volume processing efficiency are benefits of using precipitation-based methods for EV isolation [28,29]. The main disadvantages to these methods include the long processing time, retention of precipitation solvents, and contamination in the precipitate [21].

Microfluidic devices are the latest technology to be implemented for EV isolation. Small microchannels are used to separate and purify flowing samples with very high efficiency [30,31]. Either entrapment via filtration or affinity-based methods can be utilized within the microchannels to capture EVs from flowing samples [31]. Microfluidic devices allow for rapid processing, high processing efficiency, and high sample purity; however, the equipment needed, high cost, and complexity of the devices are disadvantages in using these techniques [32].

### 2.4. EV Characterization

Characterization of EVs is important for the analysis and interpretation of results from EVs. Dynamic light scattering (DLS), nanoparticle tracking analysis (NTA), nanoscale flow cytometry (nanoFACS), and transmission electron microscopy (TEM) are all effective methods to characterize EVs that provide different insights into EVs. Based on Brownian motion, DLS measures the size of the particles using a laser and measuring the light scattering by the particles. It provides an average value of relatively uniformly sized particles and is subsequently less effective for a heterogeneous solution of EVs [33]. DLS is thus best used when testing isolated samples of Exos, MVs, or ABs. NTA measures EV concentration and size distribution by using a laser beam and the Stokes–Einstein equation to measure the mean velocity and size of the particles. NTA can provide a number-based distribution of the particle-by-particle measurement. NTA is unable to distinguish between particle type and is limited in the size of EVs (1 to 1000 nm), and it is thus most suitable for samples of isolated Exos and MVs [34]. NanoFACS uses EV-specific antibodies to analyze EVs in heterogeneous samples. This method combines measurements from scattered light and fluorescent light to provide data on size, count, and distribution of EVs in the provided sample. It may use specific fluorescently labeled antibodies to stain EV surface proteins and thus determine the cell type from which the EV originated [35]. Lastly, TEM uses beams of electrons to provide a magnified, high-resolution image of the sample. TEM is useful in characterizing the morphology, size, and phenotype of EVs [36]. It may also be used to check the purity of the sample by using the image to detect the presence of non-EV particles.

## 3. Role of EVs in Diseases of the Ear, Nose, and Throat

Despite the growing interest in EVs, there have been limited studies in recent years to understand the role they may play in ENT diseases. From the research that does exist, there is a focus primarily on the role EVs play in head and neck cancers (HNCs); however, their utility and significance in various other diseases has been discovered and reported to a lesser extent. Next, we summarize the existing literature covering the role of EVs in various ENT diseases. 

### 3.1. Head and Neck Cancer

Cancer in the head and neck, primarily squamous cell carcinoma, develops in the oral cavity, nasopharynx, oropharynx, hypopharynx, and larynx [37]. Head and neck squamous cell carcinoma (HNSCC) is one of the most common causes of cancer worldwide, displaying high levels of immune suppression and poor survival rates in patients diagnosed with late-stage tumors [38,39]. Limited progress has been made in the last few decades regarding survival outcomes, and this can be most attributed to the difficulty in detecting these cancers without imaging modalities. Since HNSCC originates deep within the head and neck region, they are not easily palpable or noticeable by patients [37]. Thus, the discovery of reliable and non-invasive biomarkers is necessary to detect these cancers earlier which can lead to drastically improved patient outcomes. EVs could serve as a potential biomarker of early-stage HNSCC tumors which could easily be collected non-invasively by blood collection. Given the potential, it is no surprise many studies have been published on the role of EVs in the metastasis, lymphocyte regulation, angiogenesis, microenvironment remodeling, and drug resistance in HNSCC.

Local lymph nodes and distant metastasis is a common feature of HNC which ultimately contributes to the disease’s poor outcomes. One study investigated the role of EVs in the metastasis of nasopharyngeal cancer (NPC) cells and found that MMP13 is highly expressed in NPC cells, EVs released by NPC cells, and in the plasma of patients with NPC cells [40]. Matrix metalloproteins (MMPs) are critical enzymes in the remodeling of the extracellular matrix and lead to reduced cell adhesion and promoted invasion and metastasis [41]. An additional study found that plasma EVs containing overexpressed MMP13 may have a role in the mediation of tumor migration and invasion by upregulating Vimentin and HIF-1α expression, and reducing E-cadherin expression [42]. Latent membrane protein 1 (LMP1) is an oncoprotein that is encoded by the Epstein–Barr virus (EBV) and is a known driver of NPC [37]. It has been reported that LMP1 can promote its own packaging into CD63+/HSP70+ EVs, which is likely an important mechanism for LMP1 to engage in intracellular signal exchange and promote tumor growth [43,44]. Furthermore, CD63+/CD81+/HSP70+ EVs containing LMP1 derived, in vitro, from NPC cells have been reported to cause radio-resistance to recipient NPC cells [45]. The relationship between LMP1 and EVs is quite complex, but it can be concluded from the findings that LMP1-rich EVs possess pro-tumorigenic abilities by inducing cell proliferation, invasion, and radio-resistance in NPC [37]. An additional study found that EVs isolated from the saliva of patients with oral cavity squamous cell carcinoma and tongue cancer highly expressed miR-24-3p. This study continued to delineate that these EVs enriched with miR-24-3p stimulated non-specific cell proliferation, likely contributing to disease progression [46]. One study analyzed the proteomic profile of EVs in the same cancer cell line (HSC-3) and found the overexpression of multiple oncogenic proteins, specifically EpCAM, EGFR, and HSP90 [47,48]. These EV-packaged oncogenes certainly contribute to the promotion of tumor growth and metastasis in the setting of oral cavity squamous cell carcinoma and tongue cancer. 

HNCs are characterized as “hot tumors”, meaning they show immense infiltration of lymphocytes, macrophages, and additional active immune cells [49]. However, the immunosuppressive nature of HNCs prevents a coordinated immune response in these tumors. In vitro, it has been reported that HNC cell-derived EVs contribute to the reprogramming of immune cells. In the plasma of patients with active HNC, TSG101+ EVs induce significantly more apoptosis of CD8+ T cells, more inhibition of T cell proliferation, and increased upregulation of innate suppressive functions of CD4 + CD39+ T regulatory cells when compared with EVs of healthy patients [50]. Furthermore, an additional in vitro study found that EVs derived from oral cavity cells (Tu167) and additional HNC cells (SCC0209 and HN60) induced an immunosuppressive phenotype in CD8+ T cells [51]. Another in vitro study by Li et al. found that EVs released by hypoxic oral squamous cell carcinoma cells induce a pro-metastatic phenotype in surrounding cells via delivery of EV-miR-21 [52]. A subsequent report by the same research team found that delivery of miR-21 by CD61+/CD81+ EVs also downregulated PTEN, increased PD-L1 expression, and induced immunosuppressive activity of myeloid-derived stem cells [53]. The multiple functions of EV-miR-21 in inducing an immunosuppressive environment in HNC may make it a strong target of therapeutic interest. HPV16 infection is strongly associated with tumors of the oropharynx. Interestingly, studies have found that EVs play a differential role on immune cells depending on whether the tumor is HPV positive or HPV negative [54]. More specifically, they found that HSP70+/TSG101+ EVs from HPV-negative oral cancer (PCI-13 and PCI-30) cells reduced dendritic cell maturation while EVs from HPV-positive oral cancer (UMSCC2, UMSCC47, SCC90) cells upregulated maturation of dendritic cells, which contributes to improved outcomes of HPV-positive HNC patients [54]. In agreement, Ludwig et al. found that HPV-positive EVs were enriched with CD47 and CD247, two immune cell-related effectors, while HPV-negative EVs were enriched with MUC-1 and HLA-DA, two tumor-protective and growth-promoting antigens [50]. In the setting of immune checkpoint inhibitors, one recent study found that PD-L1+ plasma EVs reduced CD69 expression on activated T cells in HNSCC patients, offering another immunosuppressive mechanism [55]. This report further found PD-L1+ EVs in plasma of HNSCC patients was directly correlated with evidence of advanced disease, high tumor stage, and local lymph node involvement [55]. 

Multiple studies have reported the role of EV uptake in inducing upregulation of angiogenesis-related molecules resulting in the promotion of endothelial cell proliferation and migration [56,57,58]. One recent study uncovered that miR-142-3p selectively packaged into small EVs (sEVs) plays a role in enhanced angiogenesis and vascular density, both in vivo and in vitro [59]. In HNCs, multiple EV-miRNAs have been reported to play a role in regulating angiogenesis, specifically miR-23a, miR-17-5p, and miR-9 [60,61,62]. Selectively packaged proteins in EVs have also been shown to play a role in angiogenesis. One study found that HNC (OSC19, SCC61, and Detroit 562) cell-derived sEVs influence reverse signaling of Ephrin-B in endothelial cells, which promotes angiogenesis [63]. Additional studies have uncovered that CD61+/CD9+/Flotillin-1+ EVs from NPC (CNE1, CNE2, 6-18B, 5-8F) cells are enriched with 6-Phosphofructo 2-kinase/fructose 2, 6-bisphosphatase 3, a critical glycolysis-regulatory enzyme which is known to promote angiogenesis and various other pro-tumor functions [64]. Furthermore, another recent study reported that CD9+/TSG1-1+ EVs derived from HPV-positive oral cancer (UMSCC47) also promoted angiogenesis by a direct endothelial interaction and indirectly by upregulating adenosine and the A2B receptor pathway in macrophages [65]. These findings all suggest various mechanisms of EV-mediated angiogenesis in HNC. 

The tumor microenvironment is made up of a wide range of cells and proteins including tumor-associated macrophages (TAMs), immune cells, vascular cells, cancer-associated fibroblasts (CAFs), extracellular matrix, and more [66]. All of these components making up the tumor microenvironment contribute to tumorigenesis [66]. CAFs are abundantly found in the tumor microenvironment and are correlated with poor outcomes in HNC [67]. One study looking into CAFs found that NPC EVs enriched with LMP1 played a role in differentiating fibroblasts into CAFs [68]. This is a novel example of LMP1+ EVs inducing autophagy and metabolic switching of fibroblasts into CAFs, ultimately promoting proliferation and migration of NPC cells [68]. Furthermore, another study found that CD63+/CD81+/TSG101+ EVs generated by adenoid cystic carcinoma (SACC-81) cells were internalized by periodontal ligament fibroblasts and human umbilical vein endothelial cells (HUVECs), driving a malignant phenotype in both recipient cells [69,70]. High concentrations of TAMs in the tumor microenvironment, both pro-inflammatory M1 and anti-inflammatory M2, have been associated with poor outcomes in HNCs [71]. One study found that EVs from oral tongue cancers (CAL-27 and SCC9) promoted M2 macrophage polarization, influencing a pro-tumoral environment [72]. An additional study found that EV-miR-21 generated by transformed hypopharyngeal cells (FaDu) also induced M2 macrophage polarization via suppressing PDCD4 and IL12A [73]. An investigation into the role of EVs generated by HPV+ cancer cells found that EV-driven M1 macrophage polarization may play a role in the improved outcomes seen in HPV+ HNCs. Specifically, they found that CD9+/CD63+/TSG101+ EVs from HPV-positive cells (SCC47, SCC90, SCC104) stimulated M1 macrophage polarization while EVs from HPV-negative cells (SAS, CAL-27, CAL-33) stimulated M2 macrophage polarization [74]. These promising findings require further investigation and it is likely that multiple non-EV-related mechanisms also influence favorable outcomes in HPV+ HNCs. Finally, another recent report found that Alix+/TSG101+ EVs released by naive macrophages promoted migration of laryngeal cancer cells (BICR18) and induced PD-L1 expression, ultimately creating an immunosuppressive tumor microenvironment [75]. The tumor microenvironment is a complex environment that likely includes many EV- and non-EV-related regulatory mechanisms, all of which are important to further investigate. 

A primary cause of treatment failure in patients with HNCs is drug resistance. Various mechanisms can influence drug resistance, including epigenetic modifications, DNA damage repair, drug efflux, cell death inhibition, drug inactivation, and more [76]. Two studies delineated a potential role of EVs in acquired HNC drug resistance, specifically finding that inhibition of EV secretion led to a significant increase in cisplatin concentration in cisplatin-resistant oral cancer cells (H314, HSC-3-R, SCC-9-R) [77,78]. One proposed mechanism was potential EV-miR-21 targeting of programmed cell death factor 4 (PDCD4) and phosphatase tensin homolog (PTEN) [78]. Furthermore, a more recent study found that EV-miR-30a may facilitate cisplatin resistance by the upregulation of Beclin1, a known autophagy-related gene [79]. A separate study uncovered an additional factor related to cisplatin resistance, specifically the transmission of EV-miR-155 [80]. The importance of further characterization of these EVs related to cisplatin resistance must be noted; however, early studies suggest they are a critical factor in resistance. 

The roles of EVs on the metastasis and proliferation, lymphocyte regulation, angiogenesis, microenvironment remodeling, and drug resistance in HNCs are summarized in Table 1. 

### 3.2. Otitis Media 

In general, acute otitis media (OM) is a dysfunction of the eustachian tube resulting from an upper respiratory infection [81]. Diagnosis is made based on clinical examination findings, specifically physical evidence of middle ear inflammation, presence of middle ear effusion, and symptoms which include pain, irritability, and fever [81]. Chronic otitis media (COM) is a result of multiple recurrent infections of the middle ear and is often characterized by consistent middle ear effusion (MEE), which is most commonly mucoid [82,83]. This MEE is mostly made up of innate immune mediators from neutrophils, specifically neutrophil extracellular traps (NETs) and secretory mucin glycoproteins [84,85]. In vitro, OM models have shown that disease-related bacterial products drive differential expression of miRNAs within human middle ear epithelial cells (HMEECs) [86]. One study in 2017 analyzed the exosomal miRNA expression profile in middle ear effusions collected from patients with COM [87]. The research team found that exosomes collected from COM patients had a high abundance of miRNAs, most significantly being miR-223-3p, miR-451a, miR-16a-2p, miR-320e, and miR-25-3p [87]. Pathway analysis predicted these miRNAs to regulate 442 target genes; most notable were genes that upregulated many IL-8-mediated cellular functions, CXCR1/2-mediated signaling which is associated with antimicrobial defense, chemoattraction of monocytes/lymphocytes, and NF-kB pro-inflammatory upregulation resulting in angiogenesis and inflammation [87]. This was the first report of exosomal miRNAs in MEEs.

### 3.3. Chronic Rhinosinusitis

Chronic rhinosinusitis (CRS) is a persistent inflammatory disease of the paranasal sinus mucosa with multifactorial etiology, including genetic, environmental, bacterial, and immunological contributions [88]. Traditionally, CRS was classified phenotypically as occurring with (CRSwNP) or without nasal polyps, which failed to account for the range of mechanisms which can cause the disease [89]. The European Position Paper on Rhinosinusitis and Nasal Polyps 2020 advises against phenotypic classification and instead focuses on the pathophysiology of CRS to classify [90]. Given the simplicity in obtaining a clinical sample from CRS patients via nasal fluids, EVs pose a potentially useful diagnostic tool in categorizing CRS and quickly determining the most effective treatment option. Few studies have tested the cargo profile of EVs isolated from nasal fluids of CRS patients. Proteomic analysis of these samples was found in one study to show 123 differentially expressed proteins, which played a role in over 40 dysregulated signaling pathways [91]. Furthermore, significant differences in EV-cargo protein profiles were found between CRSwNP and control individuals, specifically molecular markers of CRSwNP including cystatin, glycoprotein VI, and peroxiredoxin-5 [91]. An additional report found elevated levels of cystatin-1 and -2, both epithelial protease inhibitors, in EVs isolated from nasal fluids of CRS patients [92]. These findings suggest both cystatin-1 and -2 may serve as markers of CRS with the ability to potentially predict disease phenotype as well. Another study found EVs from CRS patients to potentially lead to the formation of polyps via their role in upregulating pappalysin and serpins [93]. These findings provide another potential biomarker for diagnosis and future target for treatment of CRSwNP. 

### 3.4. Acquired Cholesteatoma 

Acquired cholesteatoma is a chronic inflammatory disease that involves an overgrowth of hyperkeratinized squamous epithelium and erosion of the bone in the middle ear [94]. Nearly all cases of acquired cholesteatoma are a result of chronic infections, with 85% of cases being bacterial infections, most often from *Pseudomonas aeruginosa* [95]. Acquired cholesteatoma ultimately causes issues due to the erosion of body structures in the middle ear, which results in breakdown of the ossicular chain and otic capsule and subsequent hearing loss, facial paralysis, vestibular dysfunction, and intracranial complications [94]. Surgical intervention by tympanomastoid surgery to remove the lesion is the only effective treatment; however, bone loss and recurrence are unavoidable, and 70% of patients require follow-up surgery within 10 years [96]. Unfortunately, repeat surgeries for acquired cholesteatoma oftentimes results in further hearing loss [97]. Inadequate treatment options highlight the importance of further understanding acquired cholesteatoma and discovery of novel targets to improve patient outcomes. One study focusing on the role of exosomes in this disease found that exosomal miR-17 of keratinocyte origin led to the upregulation of fibroblast protein expression [98]. This upregulation promotes the differentiation of osteoclasts leading to bone destruction, a hallmark of acquired cholesteatoma [98]. Furthermore, studies into the pathogenesis of this disease are required; however, this report offers a novel therapeutic target and further studies should seek to identify additional exosomal miRNAs involved in disease processes. 

### 3.5. Ototoxicity

Hearing loss, which is most often the result of cell death of the sensory hair cells of the inner ear, is a major problem across the world, affecting nearly 6.1% of the global population [99]. Sensory hair cell death can result from a variety of stressors, including noise trauma, aging, and treatment with platinum-based cancer therapeutics or aminoglycoside antibiotics, referred to as ototoxic drugs [99]. Permanent hearing loss due to ototoxic drugs affects nearly 500,000 people every year in the United States alone [100]. Uncovering the mechanisms underlying hearing loss induced by ototoxic drugs is critical to developing therapies that can avoid this therapeutic consequence. The upregulation of heat shock proteins (HSPs), specifically Hsp70, has been shown to protect sensory hair cells from aminoglycoside-induced cell death [101]. One report found that in response to heat stress, cells of the inner ear release exosomes carrying Hsp70 which then interacts with TLR4 on the hair cells [102]. These isolated exosomes provided a protective effect, thereby improving survival of hair cells treated with aminoglycoside antibiotics [102]. Further proving this mechanism, the report also found that the protective benefit was removed when these exosomes carrying Hsp70 were depleted [102]. This mechanism clearly highlights a potential therapeutic use of EVs in the setting of aminoglycoside antibiotic-induced deafness. Another report analyzed the proteomic profile of EVs isolated from mice treated with cisplatin, a platinum-based therapeutic [99]. Compared with the control, inner ear EVs from the cisplatin-treated group were found to be reduced in number and have significantly lower cargo protein concentration [99]. However, proteomic analysis showed a significant increase in protein expression of Tmem 33, Pgm1, Eif3f, Rps24, Cct8, Hsd17b4, Aldh3a1, Ddost, Aldh3a1, Eif3c, Luc7l2, and Acadvl—proteins with known roles in hearing loss [99]. Using an ischemia and reperfusion model in C57BL/6 mice, a study by Hao et al. discovered that delivery of NPC-EVs transfected with miR-21 to these mice reduced caspase-3/parvalbumin expression, increased IL-10 expression, and prevented an increase in TNF-α and IL-1β expression of cochlear hair cells [103]. This suggests that delivery of EV-miR-21 could serve to improve outcomes of cochlear damage related to ischemia [103]. Another in vivo report found that, compared with normoxic bone marrow mesenchymal stem cells (BMSCs), hypoxic BMSCs reduced cisplatin-induced ototoxicity by upregulating HIF-1α, superoxide dismutase 1 (SOD1), and SOD2 expression [104]. These findings suggest hypoxic preconditioning may offer protective effects to cisplatin-induced ototoxicity [104]. An additional in vivo report studied the effect of delivery of exosomes produced by BMSCs pretreated with heat shock (HS-BMSC-Exos) which were delivered to cisplatin-injected C57BL/6 mice [105]. This report found that trans-tympanic delivery of these HS-BMSC-Exos to cisplatin-dosed mice, reversed the upregulation of IL-1β, IL-6, TNF-α, NLRP3, ASC, cleaved caspase-1, and pro-caspase-1, thereby resulting in improved auditory sensitivity and reduced inner ear hair cell death [105]. Lastly, an in vitro investigation of gentamicin-induced ototoxicity found that delivery of Exo-miR-182-5p from mouse inner ear cells (IECs) to gentamicin-treated HEI-OC1 cells resulted in increased Bcl-2 expression and decreased FOXO3 and Bax expression [106]. These findings offer a protective mechanism of IEC-Exos in gentamicin-induced HEI-OC1 cell death [106]. 

The roles of EVs in Otitis Media, Chronic Rhinosinusitis, Acquired Cholesteatoma, and Ototoxicity are summarized in Table 2.

## 4. Summary and Future Perspectives

Although limited studies have been done thus far, reports show that EVs play significant roles in various ENT diseases. Continued studies on the characterization of EVs released in ENT diseases will allow the evolution of novel biomarkers for clinicians to use in diagnosing these diseases and therapeutic targets worthy of exploiting in drug development. Furthermore, with a more complete understanding of how EVs selectively deliver their cargo to target cells, EVs can be leveraged as a tool for drug delivery in various ENT diseases. However, before any of this can be achieved, the development of consistent and efficient methods of EV isolation from biological fluids is necessary. In summary, advances in the field of EVs, specifically on their role in ENT diseases, will propel the field and unlock a new set of tools for clinicians in the field of otolaryngology. 

## Figures and Tables

**Figure 1 medsci-11-00006-f001:**
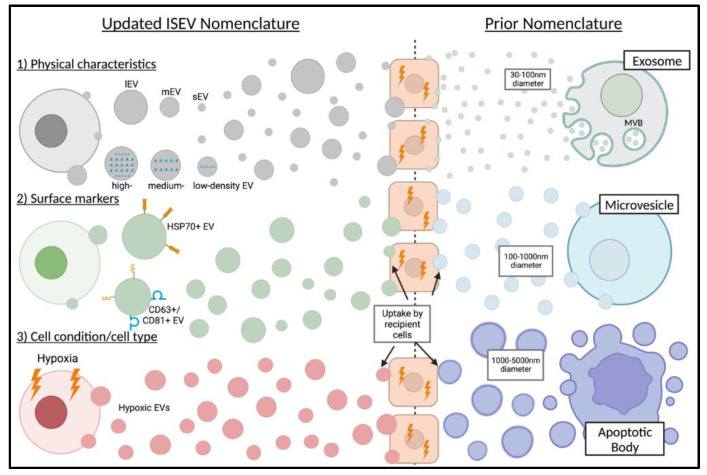
Schematic depicting both prior and updated EV nomenclature. The left side of the schematic depicts the updated 2018 ISEV guidelines and the right side depicts previously established, and still often used, EV nomenclature. EVs are released by their respective cell of origin and taken up by the recipient cells. Historically, EVs were categorized as exosomes (Exos), microvesicles (MVs), and apoptotic bodies (ABs). Updated guidelines established by the ISEV suggest categorizing EVs based on: 1) physical characteristics such as size (small, medium, or large EVs) or density (low, middle, or high density), 2) surface markers or biochemical compositions (CD63+/CD81+ EVs, HSP70+ EVs, etc.), or 3) cell condition or origin cell type (i.e., hypoxic EVs). EV—extracellular vesicle, lEV—large EV, mEV—medium EV, sEV—small EV, and MBV—membrane-bound nanovesicles.

**Table 1 medsci-11-00006-t001:** Summary of role of EVs in Head and Neck Cancers.

Reported Role of EVs	References
Metastasis and Proliferation: EVs released by NPC cells play a role in tumor migration/invasion by upregulating Vimentin and HIF-1α expression and reducing E-cadherin expression. CD63+/CD81+/HSP70+ EVs containing LMP1 cause radio-resistance in recipient NPC cells; EV-miR-24-3p isolated from oral cavity squamous cell carcinoma and tongue cancer patient saliva stimulates non-specific cell proliferation and contributes to disease progression. HSC-3 derived EVs overexpress multiple oncogenic proteins, specifically EpCAM, EGFR, and HSP90, promoting tumor growth and metastasis.	[37,40,42,43,44,45,46,47,48]
Lymphocyte regulation: TSG101+ EVs isolated from HNC patient plasma induces apoptosis of CD8+ T cells, inhibits T cell proliferation, and upregulates innate suppressive functions of CD4+CD39+ T regulatory cells. EVs derived from Tu167, SCC0209, and HN60 cells induce immunosuppressive CD8+ T cell phenotype. Delivery of miR-21 by CD61+/CD81+ EVs downregulated PTEN, increased PD-L1 expression, and induced immunosuppressive activity of myeloid-derived stem cells. HSP70+/TSG101+ EVs from HPV negative oral cancer (PCI-13 and PCI-30) cells reduced dendritic cell maturation while EVs from HPV positive oral cancer (UMSCC2, UMSCC47, SCC90) cells upregulated maturation of dendritic cells. HPV positive EVs were enriched with CD47 and CD247, two immune-cell related effectors, while HPV negative EVs were enriched with MUC-1 and HLA-DA. PD-L1+ plasma EVs reduced CD69 expression on activated T cells in HNSCC patients.	[50,51,53,54,55]
Angiogenesis: sEV miR-142-3p enhances angiogenesis and vascular density, both in vivo and in vitro. MiR-23a, miR-17-5p, and miR-9 regulate angiogenesis in HNC. HNC (OSC19, SCC61, and Detroit 562) cell-derived sEVs influence reverse signaling of Ephrin-B in endothelial cells, promoting angiogenesis. CD61+/CD9+/Flotillin-1+ EVs from NPC (CNE1, CNE2, 6-18B, 5-8F) cells are enriched with 6-Phosphofructo 2-kinase/fructose 2, 6-bisphosphatase 3, promoting angiogenesis and various added pro-tumor functions. CD9+/TSG1-1+ EVs derived from HPV positive oral cancer (UMSCC47) promote angiogenesis by a direct endothelial interaction and indirectly by upregulating adenosine and the A2B receptor pathway in macrophages.	[59,60,61,62,63,64,65]
Microenvironment remodeling: NPC EVs enriched with LMP1 influence fibroblasts differentiation into CAFs. CD63+/CD81+/TSG101+ EVs generated by adenoid cystic carcinoma (SACC-81) cells drive a malignant phenotype in periodontal ligament fibroblasts and human umbilical vein endothelial cells (HUVECs). EVs from oral tongue cancers (CAL-27 and SCC9) promote M2 macrophage polarization. EV-miR-21 released by transformed hypopharyngeal cells (FaDu) induces M2 macrophage polarization via suppressing PDCD4 and IL12A. CD9+/CD63+/TSG101+ EVs from HPV positive cells (SCC47, SCC90, SCC104) stimulate M1 macrophage polarization while EVs from HPV negative cells (SAS, CAL-27, CAL-33) stimulate M2 macrophage polarization. Alix+/TSG101+ EVs released by naive macrophages promote migration of laryngeal cancer cells (BICR18) and induce PD-L1 expression.	[68,69,70,72,73,74,75]
Drug resistance: EV secretion increases cisplatin concentration in cisplatin-resistant oral cancer cells (H314, HSC-3-R, SCC-9-R) via EV-miR-21 targeting of programmed cell death factor 4 (PDCD4) and phosphatase tensin homolog (PTEN). EV-miR-30a may facilitate cisplatin resistance by the upregulation of Beclin1, a known autophagy-related gene. EV-miR-155 also contributes to cisplatin resistance in HNC.	[77,78,79]

**Table 2 medsci-11-00006-t002:** Summary of role of EVs in Otitis Media, Chronic Rhinosinusitis, Acquired Cholesteatoma, and Ototoxicity.

Disease	Reported Role of EVs	References
Otitis Media	Disease-related bacterial products drive differential expression of miRNAs within human middle ear epithelial cells (HMEECs). Exosomes collected from COM patients have a high abundance of miRNAs, most significantly miR-223-3p, miR-451a, miR-16a-2p, miR-320e, and miR-25-3p; and these miRNAs target genes that upregulate many IL-8-mediated cellular functions and CXCR1/2-mediated signaling.	[86,87]
Chronic Rhinosinusitis	Significant differences in EV-cargo protein profiles were found between CRSwNP and control individuals, specifically molecular markers of CRSwNP including cystatin, glycoprotein VI, and peroxiredoxin-5. Elevated levels of epithelial protease inhibitors cystatin-1 and -2 are found in EVs isolated from nasal fluids of CRS patients. EVs from CRS patients potentially lead to the formation of polyps via their role in upregulating pappalysin and serpins.	[91,92,93]
Acquired Cholesteatoma	Exosomal miR-17 of keratinocyte origin leads to the upregulation of fibroblast protein expression.	[98]
Ototoxicity	In response to heat stress, cells of the inner ear release exosomes carrying Hsp70 which interacts with TLR4 on the hair cells, thereby providing a protective effect from aminoglycoside antibiotics. In mice treated with cisplatin, inner ear EVs are reduced in number and have significantly lower cargo protein concentration; proteomic analysis shows a significant increase in protein expression of Tmem 33, Pgm1, Eif3f, Rps24, Cct8, Hsd17b4, Aldh3a1, Ddost, Aldh3a1, Eif3c, Luc7l2, and Acadvl. Delivery of NPC-EVs transfected with miR-21 in vivo reduces caspase-3/parvalbumin expression, increases IL-10 expression, and prevents an increase in TNF-α and IL-1β expression of cochlear hair cells. Hypoxic BMSCs reduce cisplatin-induced ototoxicity by upregulating HIF-1α, superoxide dismutase 1 (SOD1), and SOD2 expression. Trans-tympanic delivery of HS-BMSC-Exos reverses cisplatin-induced upregulation of IL-1β, IL-6, TNF-α, NLRP3, ASC, cleaved caspase-1, and pro-caspase-1. Delivery of IEC-Exo-miR-182-5p to gentamicin-treated HEI-OC1 cells increases Bcl-2 expression and decreases FOXO3 and Bax expression.	[99,102,103,104,105,106]

## Data Availability

Not applicable.

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
