# Peer review of "The Role of Extracellular Vesicles in Diseases of the Ear, Nose, and Throat"

_medsci, 2022, doi:10.3390/medsci11010006_

Round 1
Reviewer 1 Report
In this review, the authors describe the biology and roles of extracellular vesicles focusing on the diseases of otolaryngology. The article is scientifically sound and well written. However, some points should be improved as below.
Major points:
1. Figure 1 does not provide any meaningful information to the readers. At least, there should be information about the size and characteristics of each type of extracellular vesicles.
2. Figure 1 should rather have a summary of EV classification according to the recently updated ISEV guidelines.
3. The content up to section 2.4 seems unnecessary as it is already well organized in other review papers [https://doi.org/10.3390/jnt2040013, https://doi.org/10.3390/ijms21114072]. Parts that overlap with other reviews and do not provide new information should be reduced.
4. On page 5, line 221-225, how does miR-21-mediated downregulation of PTEN and upregulation of PD-L1 relate to 'hypoxic development of the tumor microenvironment'?
5. Table 1 is merely a summary of the main text in short sentences. It would be better to replace it with illustrations
6. In section 2.10, several papers related to the exosome and ototoxicity were missing. [https://doi.org/10.1021/acschemneuro.2c00234, https://doi.org/10.3390/jcm11164743, https://doi.org/10.3390/ohbm2020006, https://doi.org/10.1186/s13036-022-00304-w, https://doi.org/10.1002/term.3089]
Minor points:
1. Section 2.6 should be renamed to Section 2.5.1 and put into Section 2.5. Same goes for the rest of the sections after 2.6.
2. On page 5, line 193-195, what is meant by 'stromal cell interaction' requires further clarification.
3. For abbreviations such as sEV, it is recommended to include the full name as well when first mentioned.
Author Response
Reviewer 1
Major points:
- Figure 1 does not provide any meaningful information to the readers. At least, there should be information about the size and characteristics of each type of extracellular vesicles.
We appreciate the reviewer’s feedback. Based on feedback we’ve created a brand new Figure 1 and legend which is drastically improved and incorporates the advice from all reviewers. We believe this figure greatly improves the manuscript and hope you will agree too.
- Figure 1 should rather have a summary of EV classification according to the recently updated ISEV guidelines.
We appreciate the reviewer’s feedback. Based on feedback we’ve created a brand new Figure 1 and legend which is drastically improved and incorporates the advice from all reviewers. We believe this figure greatly improves the manuscript and hope you will agree too.
- The content up to section 2.4 seems unnecessary as it is already well organized in other review papers [https://doi.org/10.3390/jnt2040013, https://doi.org/10.3390/ijms21114072]. Parts that overlap with other reviews and do not provide new information should be reduced.
We appreciate the reviewer’s feedback. We believe these other reviews mentioned, in addition to any other review paper on EVs in the field of ENT, greatly lack in detail on these methods compared to our current report. Given the limited reviews on EVs in ENT, we assume most readers in the field are not completely fluent in EV knowledge and information, thus in planning this paper we felt it was necessary to offer an in-depth introduction into EVs to introduce the topic to new readers. If the paper was in another field with more extensive reports on EVs, such as cancer or pulmonary, then it would make more sense to shorten this section and remove overlap. We hope you understand our explanation on why we’d like to keep this section intact.
- On page 5, line 221-225, how does miR-21-mediated downregulation of PTEN and upregulation of PD-L1 relate to 'hypoxic development of the tumor microenvironment'?
We appreciate the reviewer’s feedback. We’ve clarified these statements and expanded on them to better explain what the writer was attempting to get across. Table 1 was edited to reflect these changes as well.
- Table 1 is merely a summary of the main text in short sentences. It would be better to replace it with illustrations
We appreciate the reviewer’s feedback. We can see where the reviewer is coming from with this comment, however given the extent of variability between reported mechanisms of EVs in each disease type, it would be extremely difficult to create illustrations or schematics to completely replace Table 1. The purpose of Table 1 is to create a quick reference for readers to find the role of EVs in each disease without needing to review the paper in its entirety. To still achieve that purpose while keeping this comment in mind, we have shortened the sentences in Table 1 as much as we believe is possible to improve it. We hope this will suffice and greatly appreciate the input from Reviewer 1.
- In section 2.10, several papers related to the exosome and ototoxicity were missing. [https://doi.org/10.1021/acschemneuro.2c00234, https://doi.org/10.3390/jcm11164743, https://doi.org/10.3390/ohbm2020006, https://doi.org/10.1186/s13036-022-00304-w, https://doi.org/10.1002/term.3089]
We appreciate the reviewer’s feedback. We apologize for missing these papers in our original literature review and are thankful you informed us on them. We’ve summarized the findings of these reports and included it in our review. Thank you!
Minor points:
- Section 2.6 should be renamed to Section 2.5.1 and put into Section 2.5. Same goes for the rest of the sections after 2.6.
We appreciate the reviewer’s feedback. We’ve edited the numbering of each section as advised.
- On page 5, line 193-195, what is meant by 'stromal cell interaction' requires further clarification.
We appreciate the reviewer’s feedback. We’ve further expanded on this sentence to better explain the mechanism. These changes are reflected in Table 1 as well.
- For abbreviations such as sEV, it is recommended to include the full name as well when first mentioned.
We appreciate the reviewer’s feedback. We’ve now included the full name with the abbreviation in the first mentioning of sEVs.
Reviewer 2 Report
Here, Carnino et al., have introduced extracellular vesicles (EVs) and their significance of extracellular vesicles (EVs) in diseases of the ear, nose, and throat. This manuscript is well explained though there are just minor flaws that need to be addressed.
Minor comments:
1. Introduction and conclusion are not sufficient enough and need to be explained a bit more.
2. After the 2.5 heading, there is no explanation, and 2.6 starts abruptly. Please look into this.
2. The table appears unorganized and is very confusing. Please use a proper tabular form.
3. Figure.1. Please explain the figure in the figure legend and use different color codes for microvesicles and apoptotic bodies. Also, show the fate of these EVs in the figure.
Author Response
Reviewer 2
Minor comments:
- Introduction and conclusion are not sufficient enough and need to be explained a bit more.
We appreciate the reviewer’s feedback. We’ve expanded these sections where we could to create a more sufficient introduction and conclusion.
- After the 2.5 heading, there is no explanation, and 2.6 starts abruptly. Please look into this.
We appreciate the reviewer’s feedback. We’ve added a short paragraph immediately after the 2.5 heading and before the 2.6 heading, which offers a smoother transition into the section summarizing the literature.
- The table appears unorganized and is very confusing. Please use a proper tabular form.
We appreciate the reviewer’s feedback. The table was originally submitted in tabular form; however, the Med Sci editing team converted it into the form presented to reviewers. We believe this is Med Sci’s preferred table format, but we will leave a comment with the editing team about this. You’ll find additional improvements made to Table 1 based on additional reviewer feedback, which includes shortening EV summaries.
- Figure.1. Please explain the figure in the figure legend and use different color codes for microvesicles and apoptotic bodies. Also, show the fate of these EVs in the figure.
We appreciate the reviewer’s feedback. Based on feedback we’ve created a brand new Figure 1 and legend which is drastically improved and incorporates the advice from all reviewers. We believe this figure greatly improves the manuscript and hope you will agree too.
Round 2
Reviewer 1 Report
The authors have adequately responded to most of my concerns. However, Table 1 needs significant improvement in terms of readability and organization of information.
Author Response
The authors have adequately responded to most of my concerns. However, Table 1 needs significant improvement in terms of readability and organization of information.
We appreciate the reviewer for taking the time to review our previous point-by-point response and offer additional constructive feedback to improve our manuscript. To rectify the loss of readability and organization, we've split our original table 1 into 2 new tables. The new table 1 summarizes the role of EVs in HNC while the new table 2 summarizes the role of EVs in the remaining ENT diseases. Given that our HNC section was so copious compared to the other sections on the role of EVs in ENT diseases, we felt this would be the best way to break the information down into smaller, more readable tables plus improve organization. In addition, we've further shortened the text in these tables.
If the reviewer has further concerns with these tables more specific advice would be helpful for us to try to improve these from their present state. Similar tables as the ones currently provided can be seen in additional publications (PMID: 34368181, PMID: 32331346) and book chapters (PMID: 35659058, Hardcover ISBN: 9780323988490), so we feel these are as improved as we could make them.
Round 3
Reviewer 1 Report
The authors have made substantial efforts to improve the manuscript and the paper is now suitable for publication in Medical Sciences.